# CD44s Induces miR-629-3p Expression in Association with Cisplatin Resistance in Head and Neck Cancer Cells

**DOI:** 10.3390/cancers12040856

**Published:** 2020-04-01

**Authors:** Junichiro Chikuda, Kurataka Otsuka, Iwao Shimomura, Kagenori Ito, Hiroaki Miyazaki, Ryou-u Takahashi, Masahiro Nagasaki, Yoshiki Mukudai, Takahiro Ochiya, Toshikazu Shimane, Tatsuo Shirota, Yusuke Yamamoto

**Affiliations:** 1Division of Cellular Signaling, National Cancer Center Research Institute, Tokyo 104-0045, Japan; chang0309@dent.showa-u.ac.jp (J.C.); kurotsuk@ncc.go.jp or ishimomu@ncc.go.jp (I.S.); kagito@ncc.go.jp (K.I.); himiyaza87@gmail.com (H.M.); tochiya@tokyo-med.ac.jp (T.O.); 2Department of Oral and Maxillofacial Surgery, Showa University School of Dentistry, Tokyo 145-8515, Japan; nagasaki.masahiro@gmail.com (M.N.); mukudai@dent.showa-u.ac.jp (Y.M.); tshirota@dent.showa-u.ac.jp (T.S.); 3R&D Division, Kewpie Corporation Sengawa Kewport, Choufu-shi, Tokyo 180-0002, Japan; 4Department of Molecular Pathology, Graduate School of Biomedical and Health Sciences, Hiroshima University, Hiroshima 734-8553, Japan; 5Department of Molecular and Cellular Medicine, Institute of Medical Science, Tokyo Medical University, Tokyo 160-0023, Japan; 6Head and Neck Oncology Center, Showa University, Tokyo 142-8555, Japan; shima-tskz@dent.showa-u.ac.jp

**Keywords:** miR-629-3p, cisplatin resistance, SAS cell, cancer stem cell, apoptosis, migration

## Abstract

Cisplatin (cis-diamminedichloroplatinum II [CDDP] ) is a well-known chemotherapeutic drug that has been used for the treatment of various types of human cancers, including head and neck cancer. Cisplatin exerts anticancer effects by causing DNA damage, replication defects, transcriptional inhibition, cell cycle arrest, and the induction of apoptosis. However, drug resistance is one of the most serious problems with cancer chemotherapy, and it causes expected therapeutic effects to not always be achieved. Here, we analyzed global microRNA (miRNA) expression in CD44 standard form (CD44s)-expressing SAS cells, and we identified miR-629-3p as being responsible for acquiring anticancer drug resistance in head and neck cancer. The introduction of miR-629-3p expression inhibited apoptotic cell death under cisplatin treatment conditions, and it promoted cell migration. Among the computationally predicted target genes of miR-629-3p, we found that a number of gene expressions were suppressed by the transfection with miR-629-3p. Using a xenografting model, we showed that miR-629-3p conferred cisplatin resistance to SAS cells. Clinically, increased miR-629-3p expression tended to be associated with decreased survival in head and neck cancer patients. In conclusion, our data suggest that the increased expression of miR-629-3p provides a mechanism of cisplatin resistance in head and neck cancer and may serve as a therapeutic target to reverse chemotherapy resistance.

## 1. Introduction

Oral cancer is a malignant tumor that is categorized as a type of head and neck cancer. In 2018, more than 150,000 individuals died from oral cancer, and more than 350,000 newly developed this type of cancer [1]. In Japan, the number of patients who develop oral and oropharyngeal cancer in recent years has been steadily increasing. The estimated death toll from the cancer in 2018 was 5500 men and 2400 women [2]. In Japan, oral cancers are the 10th highest incidence in men and the 14th highest incidence in women [2]. Treatment for oral cancer depends on the stage; in the early stages, only surgery or radiation therapy (RT) is given; in advanced stages, a combination of CDDP and RT is used as adjuvant treatment [3,4]. However, in some cases, cancer cells are resistant to CDDP, and chemotherapy does not respond. Clinically, it is an obstacle to anticancer drug treatment; anticancer drug resistance is one of the most severe problems of current cancer chemotherapy [5]. Since the oral cavity has a variety of functions, such as feeding, swallowing, and speech, it has a large influence on daily life when poor clinical outcomes occur, especially in recurrent or advanced oral cancer that arise from the development of drug resistance. Therefore, elucidating the mechanisms of drug resistance is a major topic of current cancer research in clinical practice. Cancer stem cells (CSCs) have a high drug resistance and tumorigenic potential and are thought to be closely involved in recurrence and metastasis [6]. Recent evidence has demonstrated that CSCs in oral cancer express the surface marker CD44, which has been reported to have multiple splicing variants. In addition, the expression levels of specific isoforms correlate with malignancy status [7,8].

MiRNAs are noncoding RNA molecules that are 18–24 nucleotides in length, and they regulate a variety of physiological events, including the following aspects of cancer biology: tumor initiation, drug resistance, and metastasis [9,10,11]. Notably, studies have shown that miRNAs are closely involved in regulating a variety of biological processes, such as apoptosis, cellular growth, metastasis, and chemosensitivity [12,13,14,15,16]. Previous studies have reported that oral cancer cells switch from expressing the CD44 variant form (CD44v) to expressing the standard form (CD44s) during the acquisition of CDDP resistance, which results in epithelial to mesenchymal transition (EMT) induction [17]. In this study, we sought to identify key miRNA molecules that were related to the expression of CD44s and were responsible for drug resistance in oral cancer. We established oral cancer cells overexpressing CD44s and compared the expression of miRNAs in these cells with control cells that express green fluorescent protein (GFP). The resultant miRNA profiling data were compiled into a list of differentially expressed miRNAs in CD44s-expressing oral cancer cells. Among upregulated miRNA candidates, we identified that miR-210-3p, miR-629-3p, and miR-874-3p were significantly associated with CDDP resistance. Based on in vitro and in vivo experiments, our findings link miRNA biology with drug resistance in oral cancer.

## 2. Results

### 2.1. Search for miRNAs Responsible for CDDP Resistance in Head and Neck Cancer

To identify the miRNAs responsible for FP resistance, we previously established SAS/CD44s cells, which expressed CD44s, and used SAS/GFP cells as the control cells (Figure 1A) [17]. As shown in Figure 1B, the morphology of SAS/GFP cells was cobblestone-like, and SAS/CD44s cells exhibited a spindle shape. The CDDP sensitivity of SAS/CD44s cells was clearly lower than that of SAS/GFP cells (Figure 1C). According to principal component analysis (PCA) with global miRNA expression, we found that the expression profiles of SAS/CD44s cells were obviously distinct from those of SAS/GFP cells (Figure 1D). To investigate miRNAs involved in CDDP resistance, differentially expressed miRNAs Fold Chang (FC) ≥ 2, *p* < 0.05) were identified; we found that the expression levels of 14 miRNAs were significantly upregulated, and 12 miRNAs were downregulated in SAS/CD44s cells compared to those of SAS/GFP cells (Figure 1E).

### 2.2. miR-629-3p, miR-210-3p, and miR-874-3p are Involved in CDDP Resistance

To identify miRNAs responsible for CDDP resistance, we focused on the 14 upregulated miRNAs in SAS/CD44s cells. For the functional analysis, we transiently transfected each miRNA into SAS/GFP cells and analyzed cell viability following treatment with or without 20 µM of CDDP, as illustrated in Figure 2A. To identify miRNAs associated with CDDP resistance, we compared the cell viability data with and without CDDP treatment and calculated the fold changes (with CDDP/without CDDP, Figure 2B). Then, we found that miR-210-3p, miR-629-3p, and miR-874-3p remarkably enhanced cell viability under CDDP conditions. The sensitivity of CDDP was tested in miR-210-3p-, miR-629-3P-, or miR-874-3p-transfected SAS/GFP cells, and miR-629-3p transfection exhibited the largest effect (Figure 2C). CDDP concentrations for 50% and 30% survival were as follows: miR-210-3p, 1.14 µM and 3.30 µM; miR-629-3p, 1.59 µM and 4.30 µM; miR-874-3p, 1.29 µM and 3.40 µM; and negative control (NC) miRNA, 0.78 µM and 1.39 µM, respectively (Figure 2C). In addition, we performed caspase 3/7 assays to evaluate the rates of apoptosis. The results showed that miR-629-3p transfection significantly suppressed apoptosis in SAS/GFP cells, although miR-210-3p and miR-874-3p transfection did not decrease apoptosis in SAS/GFP cells (Figure 2D). To further explore the effect of miR-210-3p, miR-629-3p, and miR-874-3p on the migration of SAS/GFP cells, we performed migration assays. All three miRNAs significantly promoted the migration of SAS-GFP cells; in particular, miR-629-3p showed the highest effect (Figure 2E). Although there was no significant difference in Kaplan–Meier survival analysis and the probability of survival was not directly related to drug resistance, we found that a higher expression of miR-629 tended to be associated with poor overall survival (Figure 2F). Furthermore, it was confirmed that higher miR-629 expression was significantly correlated with a worse prognosis in esophageal adenocarcinoma, kidney renal papillary cell carcinoma, liver hepatocellular carcinoma, and pancreatic ductal adenocarcinoma (Appendix A).

### 2.3. Confirmation of miR-629-3p Effects on CDDP Resistance in Other Head and Neck Cancer Cell Lines

Since miR-629-3p, 210-3p and miR-874-3p were confirmed to be involved in CDDP resistance in SAS cells, we tested their function with other head and neck cancer cell lines: HSC-3 cells and HSC-4 cells. Both cell lines had a cobblestone morphology (Figure 3A left and D left), and we examined the sensitivity of HSC-3 cells and HSC-4 cells to CDDP following transfection with miR-210-3p, miR-629-3P, miR-874-3p, or NC (Figure 3A right and D right). As shown in the Figure 3A and D right graphs, it was found that these miRNAs tended to influence the sensitivity to CDDP, although the effects were not identical. Additionally, caspase 3/7 activity was significantly decreased in HSC-3 and HSC-4 cells that were transfected with miR-629-3p (Figure 3B,E). Likewise, we conducted a transwell assay to test the migration of HSC-3 and HSC-4 cells after miRNA transfection. Similar to SAS cells, miR-629-3p and miR-210-3p transfections significantly promoted cell migration in both cell lines (Figure 3C,F). Collectively, these results indicate that miR-629-3p decreases CDDP sensitivity in HSC-3 and HSC-4 cells, and it is involved in the suppression of apoptosis and promotion of migration.

### 2.4. miR-629-3p Influences Multiple Pathways in Head and Neck Cancer Cells

To understand the molecular basis of the phenotypic changes in SAS/GFP cells following miRNA transfection, we performed whole transcriptome profiling of SAS/GFP cells transfected with each miRNA. PCA mapping of whole transcriptome data showed a divergence in cells transfected with each miRNA (Figure 4A). From the heatmap of differentially expressed genes (798 genes, False Discovery Rate (FDR) < 0.05), miR-629-3p transfection clearly showed distinct expression patterns (Figure 4B). When directly comparing miR-629-3p-transfected SAS/GFP cells with NC cells, we identified 969 upregulated genes (*p* < 0.05, FC ≥ 2) and 447 downregulated genes (*p* < 0.05, FC ≥ 2, Figure 4C). Compared with miR-210-3p and miR-874-3p transfection, there was a larger number of genes influenced by miR-629-3p transfection (Figure 4D). To scrutinize the phenotype of miR-629-3p-transfected SAS/GFP cells, we performed Ingenuity Pathway Analysis (IPA), which revealed the significant alteration of several molecular functions involved in cell migration (Figure 4E). Additionally, Gene Set Enrichment Analysis (GSEA) revealed the significant enrichment of gene sets for “drug metabolism other enzymes” and “epithelial to mesenchymal transition (EMT)” in miR-629-3p-transfected SAS/GFP cells when compared with NC cells (Figure 4F). In addition, some EMT markers were changed in miR-629-3p-transfected SAS/GFP cells in microarray data (Appendix A). Thus, miR-629-3p influenced a large number of genes that were presumably involved in CDDP sensitivity.

To identify genes affected by miR-629-3p, we used a public database for miRNA target prediction that combined the data from three algorithms: TargetScan, miRDB, and micro T-CDS (Figure 5A). When combining the in silico data with our microarray data, we reduced the list of miR-629-3p targets to 34 genes that were downregulated in miR-629-3p-transfected SAS/GFP cells (FC ≥ 1.75, *p* < 0.05, Figure 5B). To test the suppression of these genes by miR-629-3p, we performed qRT-PCR for the 34 genes. When the average CT value was more than 35, we excluded them because of insufficient expression levels, and then the rest of the genes were plotted (Figure 5C). This qRT-PCR data revealed that a number of genes might be influenced by the miR-629-3p transfection in SAS cells. Among these 19 genes, the expression levels of 11 genes were significantly decreased in miR-629-3p-transfected SAS cells; the suppression of two cancer-related genes (PTP4A1 and INPP5A) was also confirmed in HCS-3 and HCS-4 cells (Appendix A). Collectively, these data suggested that the differentially expressed genes by miR-629-3p orchestrate and contribute to the malignant phenotype such as drug resistance.

### 2.5. miR-629-3p Exerts a Functional Role in Drug Resistance in Vivo

To investigate the function of miR-629-3p in an in vivo xenograft model, we established stable miR-629-3p-expressing SAS-luc cells and NC SAS-luc cells with the electroporation of miR-629-3p expressing plasmid and negative control plasmid, respectively. The expression level of miR-629-3p was confirmed by qRT-PCR (Figure 6A). The cells were transplanted by injection into both flanks of 5-week-old male BALB/c nude mice (Figure 6B). Mouse weight and tumor size were assessed twice per week by comparing the CDDP administration group with the PBS administration group (Figure 6B,C). As expected, in vivo imaging revealed that the tumor size of NC xenografted mice was generally smaller than that of miR-629-3p-expressing xenografted mice in the CDDP-treated group (Figure 6D,E). Although a significant difference in tumor size was not observed in the PBS-treated group between miR-629-3p-expressing and NC SAS-luc cells, we observed significant changes in the CDDP group (Figure 6F). When the tumors were collected on day 25, the tumor weight in the CDDP group was significantly different when comparing tumors from miR-629-3p-expressing and NC SAS-luc cells (Figure 6G). Downregulations of putative miR-629-3p targeted genes as shown in Figure 5C were examined using the xenografed tumor samples, and many of them tended to decrease in miR-629-3p-expressing SAS cells (Appendix A). Taken together, these findings confirmed that miR-629-3p is involved in the CDDP resistance in SAS cells in vivo.

## 3. Discussion

Although there are several studies showing the relationship between drug resistance and miRNA expression in cancer, the molecular mechanisms by which tumor cells acquire drug resistance are not completely understood in head and neck cancer. In this study, we demonstrated that the expression of 14 miRNAs including miR-629-3p was upregulated in SAS/CD44s cells, which are resistant to CDDP; further, enhanced miR-629-3p expression inhibited apoptotic cell death and promoted cellular migration. Based on bioinformatics approaches, we identified several genes whose expressions were reduced by miR-629-3p. Importantly, miR-629-3p influenced drug sensitivities in several types of oral carcinoma cell lines such as SAS, HSC-3, and HSC-4, and a high expression of miR-629-3p positively correlated with poor survival rate in oral cancer patients, although it was not statistically significant. In addition, the effect of miR-629-3p was confirmed in a xenograft mouse model of oral cancer cells treated with CDDP. Thus, miR-629-3p is a novel miRNA that is responsible for drug resistance in oral carcinomas. 

Although miR-629-3p has been reported as oncogenic miRNAs in various cancers, there are few published studies on the role of miR-629-3p in drug resistance, and the expression level and functional role are unknown. Past studies have demonstrated that the suppression of miR-629 enhances the sensitivity of cervical cancer cells to 1′S-1′-acetoxychavicol acetate by regulating RSU1 [18]. It was also revealed that miR-629-3p-induced surfactant protein C (SFTPC) downregulation promoted cell proliferation and predicted poor survival in lung adenocarcinoma [19]. In addition, it has been previously reported that miR-629-3p may serve as a novel biomarker and potential therapeutic target for lung metastases in triple-negative breast cancer [20]. Our results showed that the upregulation of miR-629-3p enhanced sensitivity toward CDDP by reducing apoptotic cell death and increasing cell migration; this finding indicated that miR-629-3p may also play a role in regulating the response toward anticancer agents.

In addition, we also demonstrated that miR-629-3p regulates CDDP resistance by targeting several genes. Its expression was clearly downregulated in oral cancer cell lines following miR-629-3p transfection. It has been reported to contribute to CDDP resistance in urothelial cancer and 5-fluorouracil (5-FU) resistance in gastric cancer [21,22]. *FBXO32*, which is one of the candidates but not confirmed by qRT-PCR, is known to be related to drug resistance. Tanaka et al. reported that CDDP resistance was caused by EMT through FBXO32 dysregulation [21]. Presumably, this molecular mechanism, at least in part, involved in our experimental model. In this study, we showed that miR-629-3p downregulated a panel of genes that were predicted by computational analysis and microarray data. One limitation of this study is that we could not determine that these downregulated genes were miR-629-3p’s direct targets; one of the reasons is because of the lack of good antibodies for Western blots. Since one miRNA regulates hundreds of target genes, we assume that miR-629-3p orchestrates these gene expressions, leading to drug resistance. Thus, our data indicate that this tumor suppressor miRNA could also play a role in regulating sensitivity toward anticancer agents. Another limitation is that we only examined the function of miR-629-3p by overexpression both in vitro and in vivo. To further investigate the effect of miR-629-3p on drug resistance, silencing miR-629-3p with a specific inhibitor is a great option, which supports demonstrating miR-62p-3p’s role in promoting tumorigenesis and CDDP resistance. In addition, when we consider miR-629-3p as a therapeutic target, the strategy of miRNA silencing is necessary for clinical applications. 

## 4. Materials and Methods

### 4.1. Plasmids and Establishment of Stable Cell Line Expressing miR-629-3p

To generate vectors for the expression of miR-629, stem loops of miR-629 were cloned into the multi-cloning sites of a pCDNA3.1-N-eGFP expression vector (GenScript Japan, Tokyo, Japan), and expression was driven by the CMV promoter. Cells transfected with the empty vector pCDNA3.1 (GenScript) were used as a control. For establishing stable miR-629-3p-expressing SAS-luc cells and NC SAS-luc cells, these plasmids were transfected with electroporation (Nucleofector™ 2b, Lonza, Basel, Switzerland). Using Nucleofector™ 2b, transfected plasmids are frequently integrated into the genome, generating the stable expression of a target gene. To enrich stable cell lines, the transfected cells were selected with G418 at 300 µg/mL. Before using these cells for animal transplantation, the expression level of miR-629-3p was tested by qRT-PCR.

### 4.2. Cell Lines and Cell Culture

Human oral squamous carcinoma cell lines SAS, HSC-3, and HSC-4 cells were used for the cellular experiments in this study. SAS, HSC-3, and HSC-4 cells were obtained from the Japanese Collection of Research Bioresources Cell Bank. SAS cells, HSC-3, and HSC-4 cells were cultured in Dulbecco’s Modified Eagle Medium (DMEM) (Thermo Fisher SCIENTIFIC, Waltham, MA, USA) supplemented with 10% (v/v) fetal bovine serum (Thermo Fisher SCIENTIFIC, Waltham, MA, USA) and 1% (v/v) Antibiotic-Antimycotic (Thermo Fisher SCIENTIFIC, Waltham, MA, USA). All cells were maintained in a humidified incubator containing 5% CO_2_ at 37 °C. 

### 4.3. miRNA Transfections

For miRNA transfection, we transfected human oral squamous carcinoma cell lines with 25 nM of miR-mimic (Dharmacon, Lafayette, CO, USA) or a negative control (Applied Biosystems, Waltham, MA, USA) using DharmaFECT 1 Transfection Reagent (Dharmacon) according to the manufacturer’s protocol.

### 4.4. Microarray and Bioinformatics

To perform the miRNA microarray, SAS/GFP cells and SAS/CD44s cells were established as previously described, and miRNA was extracted [17]. Microarray analysis (Agilent-046064 Unrestricted_Human_miRNA_V19.0_Microarray, Tokyo, Japan) was performed according to the protocol from Agilent Technologies Inc. To perform the mRNA microarray, mRNA was extracted as follows. First, miRNAs were transfected after seeding the cells and confirming sufficient adhesion. Twenty-four hours after transfection, total RNA and miRNA were extracted from the cells using an RNeasy Mini kit (QIAGEN, Hilden, German). Microarray analysis (Agilent-072363 SurePrint G3 Human GE v3 8x60K Microarray 039494, Tokyo, Japan) was performed as described above. Raw and normalized microarray data are available in the Gene Expression Omnibus database (accession numbers GSE131166 and GSE129875). 

### 4.5. Microarray Analsysis and Bioinformatics

For GSEA, the javaGSEA 3.0 Desktop Application (http://software.broadinstitute.org/gsea/index.jsp) was used. Principal statistics were FDR<0.1, *p* < 0.05 to produce a normalized enrichment score (NES). Activated upstream regulators were considered when the IPA activation z-score value was between two- and four-fold (*p* < 0.001). For IPA, the analysis was performed following the manufacturer’s instructions (https://www.qiagenbioinformatics.com/products/ingenuity-pathway-analysis/).

### 4.6. Quantitative Reverse Transcription PCR (qRT-PCR) 

For total RNA and miRNA extraction, an RNeasy Mini Kit (Qiagen) was used, and for cDNA systhesis, a High Capacity cDNA Reverse Transcription Kit (Applied Biosystems) or a TaqMan MicroRNA Reverse Transcript Kit (Applied Biosystems, Waltham, MA, USA) was used based on the manufacturer’s protocol. The synthesized cDNA samples were applied to real-time PCR (a StepOnePlus Real-Time PCR System, Applied Biosystems, Waltham, MA, USA) with SYBR Select Master Mix (Applied Biosystems, Waltham, MA, USA). The primers are listed in Appendix A. TaqMan MicroRNA Assays were purchased from Applied Biosystems and used for qRT-PCR analyses of miRNAs (Applied Biosystems). Expression levels of mRNA and miRNA were normalized to those of β-actin or RNU6B, respectively, and the relative expression was calculated using the 2^ΔΔCt^ method.

### 4.7. Cell Proliferation Assay

Cell proliferation was analyzed using a Cell Counting Kit-8 (Dojindo Molecular Technologies, Inc., Tokyo, Japan) as described in the manufacturer’s instructions. In brief, cells were seeded at a density of 5 × 10^3^ cells/well in 96-well plates, and miRNAs were transfected after confirming sufficient adhesion. Each concentration of cisplatin was added 12 h after transfection. After another 48 h, the CCK-8 reagent was incubated with the cells for 2 h, and the absorbance was measured at 450 nm using a microplate reader (MOLECULAR DEVICES, SpectraMax® iD5, San Jose, CA, USA).

### 4.8. Apoptosis Assay

To evaluate apoptotic activity, a luminescent caspase-3/7 activation assay was performed. Oral cancer cells were seeded as described above, and 12 h after miRNA transfection, cisplatin was added at a concentration of 20 μM. After a 48-hour incubation, Caspase-Glo® reagent (Caspase-Glo® 3/7 assay; Promega, Madison, WI, USA) was added and incubated with the cells for 1 h, and then the activity of caspase-3/7 was measured using a microplate reader (MOLECULAR DEVICES, SpectraMax® iD5) according to the manufacturer’s protocol. 

### 4.9. Migration Assay

To evaluate migration ability, migration assays were carried out in transwell chambers with 8-μm pore size (Corning, New York, NY, USA). Then, 48 h after transfection, the cells (5 × 10^4^/well) in 500 μL of DMEM without FBS were added to the upper chamber, while 750 μL of DMEM with 10% FBS (Thermo Fisher SCIENTIFIC, Waltham, MA, USA) was placed in the lower chamber. The chambers were placed in 24-well plates and incubated at 37 °C with 5% CO_2_ for 48 h. After incubation, the cells on the upper surface of the filter were wiped off using a cotton swab. Subsequently, the cells on the lower surface of the filter were fixed and then stained with Diff-Quik (Scientific Products, Harleco, Gibbstown, NJ). The stained cells were computationally counted by Image J software in at least three randomly selected fields per well, and then the mean value was recorded. 

### 4.10. Animal Experiments

All animal experiments were performed in accordance with the protocol of the National Cancer Center Institutional Animal Care and Use Committee (Ethic code: #T17-034). Five-week-old male BALB/C nude mice (Charles River Laboratories, Japan) were used for the xenograft model. A total of 5 × 10^5^ stable miR-629-3p-expressing SAS-luc cells and NC SAS-luc cells (Generated in 4.1) were injected into both flanks (right and left, respectively) of the mice with 100 μL of Matrigel/PBS (50% final concentration) to establish xenograft models. Considering the right/left body difference in mice, for half of the mice, NC SAS-luc cells were injected into the right side, and stable miR-629-3p-expressing SAS-luc cells were injected into the left side; for the other half of the mice, vice versa. Six days after inoculation, each mouse was intraperitoneally administered PBS (*n* = 6 groups) or CDDP at a dose of 5 mg/kg body weight (*n* = 7 groups) once a week for 4 weeks. Mice were monitored carefully, and the size of their tumors was measured using a vernier caliper. Images were analyzed with Living Image software (Caliper Life Sciences, Waltham, MA, USA). Tumors were harvested 4 weeks after inoculation of cancer cells, and tumor weight was measured.

### 4.11. Kaplan–Meier Plot

Kaplan–Meier plots were generated using the Kaplan–Meier plotter website (miRpower for pan-cancer) from a database of public miRNA expression datasets (http://kmplot.com/ analysis). HRs and P-values (log rank P) are shown for each survival analysis.

### 4.12. Statistics

Data are presented as the mean ± s.d. of *n* = 3–4 biological samples analyzed in triplicate. For two-group comparisons, statistical significance was determined by Student’s *t*-test. For multiple comparisons, significant differences in average values were analyzed using one-way ANOVA with Tukey’s HSD or Dunnett’s post hoc test. The limit of statistical significance for all analyses was defined as **p* < 0.05.

## 5. Conclusions

Our study indicates that miR-629-3p possesses an oncogenic function related to drug resistance, and consequently, it has an important role in the development of head and neck cancer. Our results provide a strong rationale for the potential use of miR-629-3p as a therapeutic target to reverse chemotherapy resistance in tongue cancers.

## Figures and Tables

**Figure 1 cancers-12-00856-f001:**
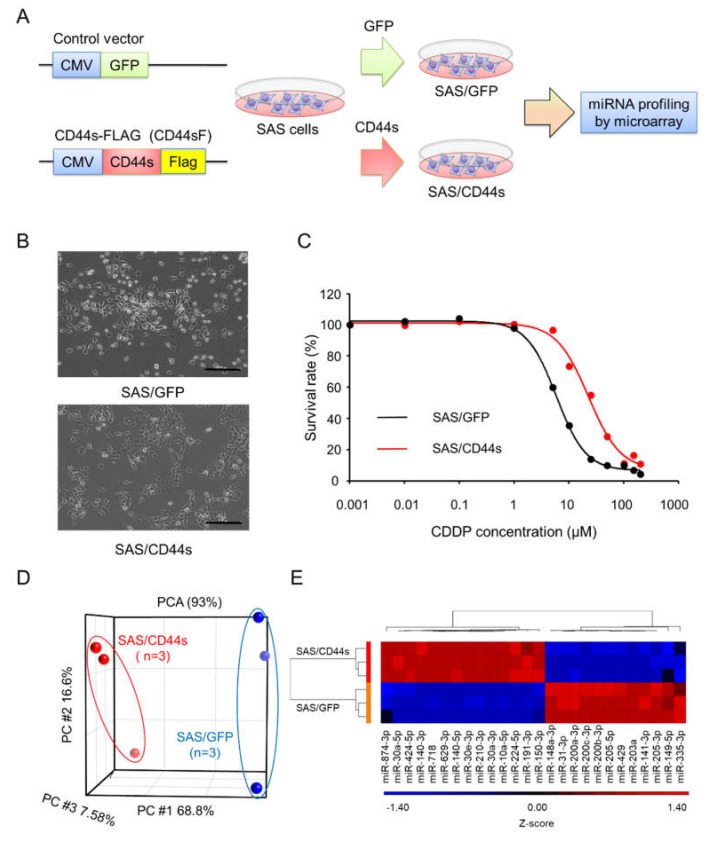
Search for microRNAs (miRNAs) responsible for cisplatin (cis-diamminedichloroplatinum II [CDDP]) resistance in head and neck cancer. (**A**) Overview of the method used to establish the CD44 standard form (CD44s)-expressing or green fluorescent protein (GFP)-expressing SAS derivatives and to perform miRNA profiling. A C-terminal Flag-tagged CD44 expression vector was used to develop CD44s-expressing SAS cells. A GFP-expressing construct was used as a negative control. (**B**) Morphologies of SAS/GFP and SAS/CD44s cells. Scale bar, 500 µm. (**C**) The sensitivity of SAS/GFP and SAS/CD44s cells to CDDP. CDDP sensitivity was determined by MTT assay. (**D**) Principal component analysis (PCA) mapping in SAS/GFP and SAS/CD44s cells. (**E**) Unsupervised clustering with a heatmap focusing on differentially expressed miRNAs in SAS/GFP and SAS/CD44s cells.

**Figure 2 cancers-12-00856-f002:**
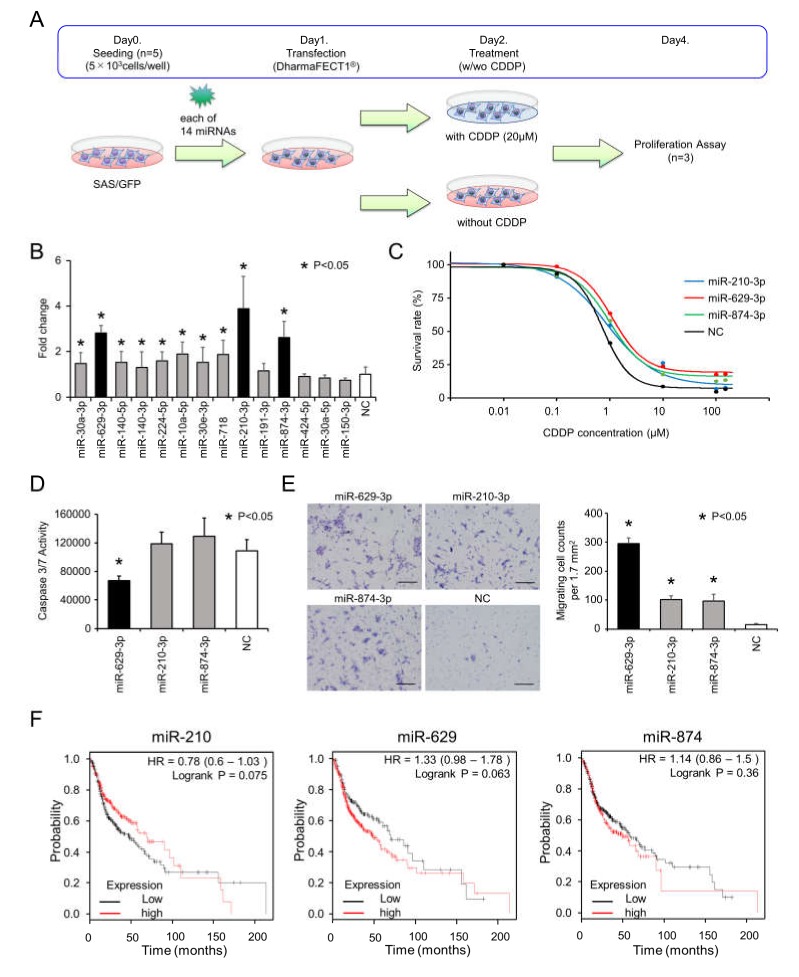
Functional analysis of miR-629-3p, miR-210-3p, and miR-874-3p for CDDP resistance. (**A**) Schematic representation of cell viability analysis with and without cisplatin. After 24 hours, medium was switched to CDDP (20 µM) condition or no CDDP condition, and after another 48 h, the Cell Counting Kit-8 (CCK-8) reagent was reacted for 2 h, and the absorbance was measured at 450 nm in a microtiter plate reader. (**B**) Cell viability of SAS cells transfected with each miRNA. Fold changes of CDDP(+)/CDDP(−) was calculated for each miRNA. Cell viability was normalized to negative control (NC). (**C**) The sensitivity of CDDP in miR-210-3p, miR-629-3p, miR-874-3p, or NC-expressing SAS cells. CDDP sensitivity was determined by MTT assay. (**D**) Caspase 3/7 activity assay of in miR-210-3p, miR-629-3p, miR-874-3p or NC-expressing SAS cells. (**E**) Representative images (left) and relevant quantification (right) of migration assays in in miR-210-3p, miR-629-3p, miR-874-3p, or NC-expressing SAS cells. Scale bar, 100 µm. (**F**) Kaplan–Meier plots of the probabilities of survival in head and neck cancer cases classified according to the expression levels of miR-210, miR-629, and miR-874.

**Figure 3 cancers-12-00856-f003:**
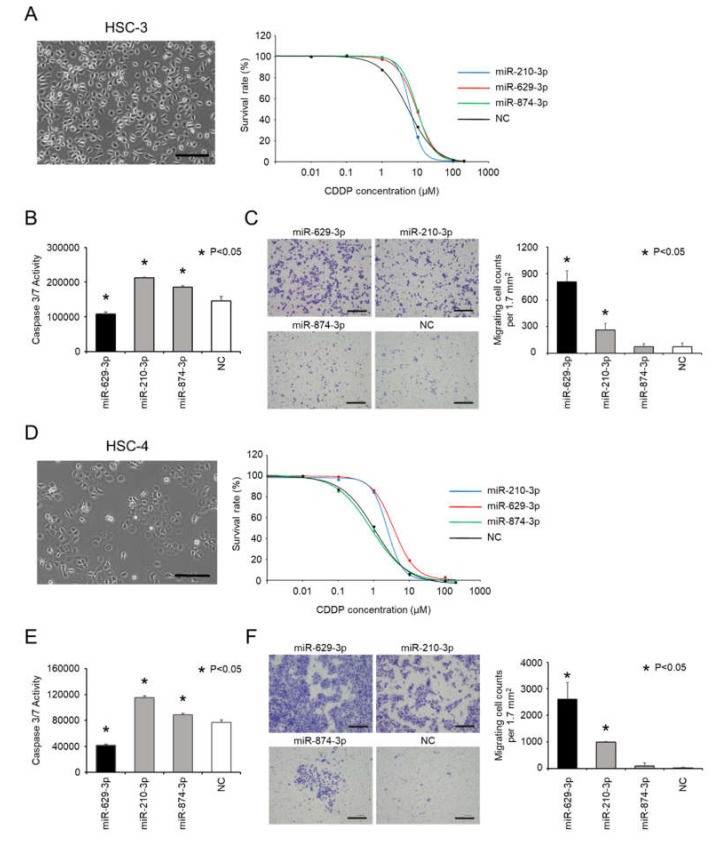
miR-629-3p induces CDDP resistance in several head and neck cancer cell lines. (**A**,**D**) Morphology of HSC-3 cells and HCS-4 (scale bar, 500 µm) and the sensitivity to CDDP in miR-210-3p-, miR-629-3p-, miR-874-3p-, or NC-expressing HCS-3 and HCS-4 cells, respectively. (**B**,**E**) Caspase 3/7 activity assay of miR-210-3p, miR-629-3p, miR-874-3p, or NC-expressing HCS-3 cells and HCS-4 cells, respectively. (**C**,**F**) Representative images (left) and relevant quantification (right) of migration assays in miR-210-3p-, miR-629-3p-, miR-874-3p-, or NC-expressing HCS-3 cells and HCS-4 cells, respectively. Scale bar, 100 µm.

**Figure 4 cancers-12-00856-f004:**
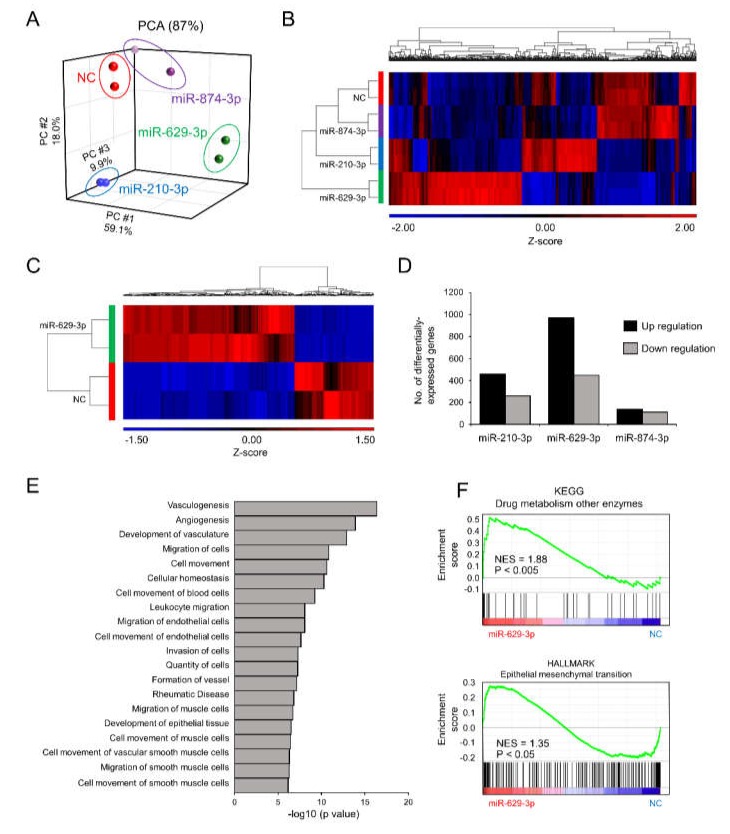
Microarray analysis reveals miR-629-3p functions in head and neck cancer cells. (**A**) PCA mapping of global gene expression of SAS/GFP cells transfected with each miRNA. (**B**) A heatmap analysis focusing on differentially expressed genes (798 genes, FDR < 0.05). (**C**) Differentially expressed genes between miR-629-3p-expressing SAS/GFP and control cells; two-fold or more and *p* < 0.05. (**D**) The number of differentially expressed genes in miR-629-3p-expressing SAS/GFP and control cells; two-fold or more and *p* < 0.05. (**E**) IPA identified canonical pathways. The log10 *p* value was plotted. (**F**) Gene set enrichment analysis (GSEA) comparing miR-629-3p-expressing SAS/GFP and control SAS cells. NES: normalized enrichment score. The *p*-value was calculated by GSEA.

**Figure 5 cancers-12-00856-f005:**
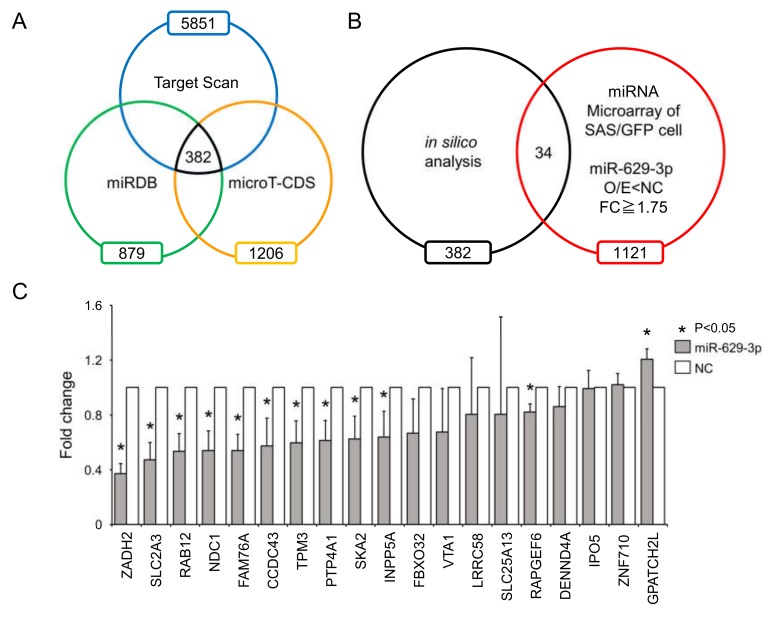
Downregulated genes after miR-629-3p transfection. (**A**) Venn diagram showing three miRNA target prediction algorithms of miR-629-3p (in silico analysis). (**B**) Comparison of in silico analysis with microarray data (downregulated genes in miR-629-3p transfection) for predicting potential miR-629-3p target genes. (**C**) Quantitative RT-PCR analysis of 19 genes in miR-629-3p-expressing SAS/GFP cells. Expression levels were normalized to β-actin, and relative expression was calculated using the comparative CT method.

**Figure 6 cancers-12-00856-f006:**
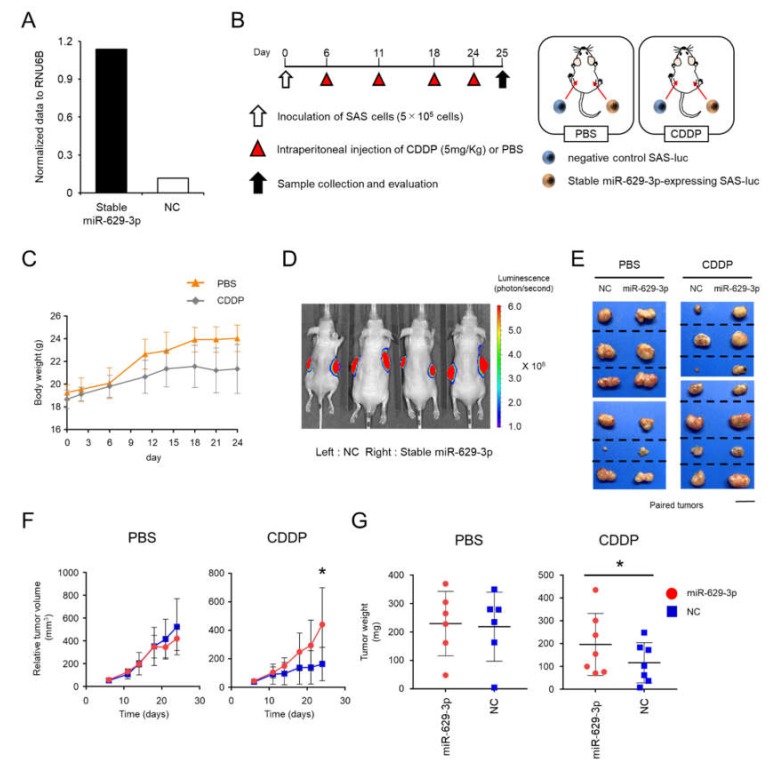
miR-629-3p exerts a functional role in drug resistance in an in vivo xenograft model. (**A**) Stable overexpression of miR-629-3p in SAS-luc cells was used for an in vivo study. A mock vector was transfected as a negative control sample. The expression levels of miR-629-3p were measured by qRT-PCR. The expression levels were normalized with RNU6B. (**B**) Experimental scheme for the in vivo study of miR-629-3p function in CDDP resistance. Stable miR-629-3p-expressing SAS cells and control SAS cells were injected into mouse flanks, respectively. (**C**) Mouse weight during CDDP treatment (CDDP: *n* = 7; PBS: *n* = 6). (**D**) Representative image of bioluminescence reveals SAS-luc cells. Control cells (left) and stable miR-629-expressing SAS cells (right). Images were taken using an in vivo imaging system (IVIS). (**E**) Representative pictures from harvested tumors. The photos showing paired tumors (xenografted tumors of stable miR-629-3p-expressing SAS-luc cells and negative control SAS-luc cells). PBS-treated group: *n* = 6 and CDDP-treated group: *n* = 7. Scale bar: 1 cm. (**F**) Tumor size. Red: stable miR-629-expressing SAS cells. Blue: negative control SAS cells. (G) Tumor size was measured twice per week. The values are the mean ± SD. PBS-treated group: *n* = 6; CDDP-treated group: *n* = 7.

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
