# Peer review of "CD44s Induces miR-629-3p Expression in Association with Cisplatin Resistance in Head and Neck Cancer Cells"

_cancers, 2020, doi:10.3390/cancers12040856_

Round 1
Reviewer 1 Report
I think the revised version was adequately modified in accordance with the reviewer's suggestion, therefore, it is now suitable for publication from Cancers.
Reviewer 2 Report
No specific comments
This manuscript is a resubmission of an earlier submission. The following is a list of the peer review reports and author responses from that submission.
Round 1
Reviewer 1 Report
This article describes miR-629-3p is related to pharmacological resistance of cisplatin to head and neck cancer cells based on the large amounts of analysis. The results required huge experimental work, long-time effort, huge costs, and bioinformatics methods. I think it would be worthy of publication from "Cancers" after minor corrections, therefore, I'd like to offer some points to be modified for constructive criticism.
- Line 73-80 should be removed. Perhaps, these sentences were defaulted in the format of manuscript.
- Figure1-B; photos of SAS/GFP and SAS/CD44s should be enlarged. The shape of SAS/GFP cells and SAS/CD442 were described as cobblestone-like and spindle shape, respectively. However, the photos of each cell are too small to be identified.
- Line 143-144 "although the effects of miRNA transfection were not identical to those observed in the SAS cells":What is the point of this sentence? Does it mean that the effects of miRNA transfection on HSC-3 cells and HSC-4 cells were different from those on SAS cells? As far as I know, these three cell-lines show quite different characteristics, although they are all head and neck cancer cell lines. So it would be just something natural. Please consider what you really wanted to put it into the words.
- Figure3-A,D: these photos also should be enlarged.
- Line 180-185, Figure 5-C, S2; To summarize these data, the expression level of cancer-related genes such as PTP4A1, INP5A and FBXO32 were suppressed by miR-629-3p transfection in HSC-3 and HSC-4 cells, but that FBXO32 was not suppressed in SAS cells, is it right? If so, please clearly describe it. From the current description makes readers think all the three genes were suppressed in each three cell line and confused when they see Figure5-c, without asterisk at FBXO32.
- Figure 6-E, 5th from the top of xenografted tumors which received PBS treatment seems to be quite small. From the left photo, readers may consider that PBS would be effective for tumor cells with one-sixth probability. Moreover, from the right photo, readers may think CDDP would be effective to xenografted tumor with two-seventh probability comparing normal cells and miR-629-3p transfected cells. It is because of the rest of the five pairs, that the size of both xenografted tumors do not seem to be different. Please clearly describe the result.
Reviewer 2 Report
- No information was provided regarding the Kaplan–Meier plots of the probabilities of survival in head and neck cancer cases. Where were the data retrieved from?
- It is suggested to examine the expression of miR-629-3p in clinical samples in order to verify the finding.
- It will be better to conduct the experiments and confirm the possible downstream factors (PTP4A1, INPP5A, and FBXO32?) that are regulated by miR-629-3p. Also, it would be good to examine the expression of the putative downstream genes in the excised tumors from in vivo study.
- In the discussion section, it was stated: “enhanced miR-629-3p expression inhibited apoptotic cell death and promoted cellular migration related to EMT phenotype”. Please specify the rationale that the EMT phenotype was confirmed in the data.
- It was mentioned in the discussion that “we also demonstrated that miR-629-3p regulates CDDP resistance by targeting several genes, including FBXO32”. However, it seemed like there was no difference in the gene expression of FBXO32 between miR-629-3p transfection and NC cells.
- The author suggested that the decrease in CDDP sensitivity by miR-629-3p was associated with suppression of apoptosis and promotion of migration/ EMT. This statement was not well elaborated. The signaling pathways involve in apoptosis and drug resistance may not be overlapped. Same goes to EMT. The authors need to avoid using some terms without proper explanation.
- Please remove the instruction from the journal in the manuscript.
Reviewer 3 Report
The authors established a cellular reporter system to identify potential miRs which are involved in CDDP resistance. The study was well designed and experiments well carried out.
However, there are only minor issues that require authors' attention.
- Since the cells express luciferase, why wouldn't the authors track tumor growth using non-invasive molecular imaging over time. It could have detected distant metastasis since according to the in vitro data that miR-629-3p promotes migration in head and neck cancer cell lines.
- Silencing miR-629-3p (using inhibitor of miR-629-3p) should have been included in the in vivo experiment as one of the experimental groups. This would have provided even stronger support for miR-62p-3p's role in promoting tumorigenesis and CDDP resistance.
- The experimental procedure for animal experiment requires a more detailed description. For instance, the type of tumor cells and the treatment of the cells prior to inoculation should have been clarified.
- The authors should provide verifying evidence that miR-629-3p mimic transfected was still present/functional in the cells inoculated. This is essential since mimic molecules transfected is transient; multiple and repeated injections are required to sustain the require concentration to function in vivo experiments (especially the experiment was as long as 25 days).
- Continuing from #4, according to the in vitro results presented in this study, it is clear that miR-629-3p is associated with anti-apoptotic function, and it would be plausible that an increased miR-629-3p level was also associated with higher colony-generating and proliferative abilities.